# Application of the Active-Fluidics System in Phacoemulsification: A Review

**DOI:** 10.3390/jcm12020611

**Published:** 2023-01-12

**Authors:** Yu Luo, Guangcan Xu, Hongyu Li, Tianju Ma, Zi Ye, Zhaohui Li

**Affiliations:** 1Medical School of Chinese People’s Liberation Army, Beijing 100853, China; 2Department of Ophthalmology, Chinese People’s Liberation Army General Hospital, Beijing 100853, China

**Keywords:** cataract, phacoemulsification, active-fluidics system, gravity-fluidics system

## Abstract

The fluidics system is an indispensable and primary component of phacoemulsification. Both the gravity-fluidics system and active-fluidics system are commonly used in practice. The irrigation pressure of the gravity-fluidics system is determined by the bottle height, which is relatively constant, while the active-fluidics system is paired with a cassette that contains pressure sensors to monitor intraocular pressure changes. The active-fluidics system allows surgeons to preset a target intraocular pressure value, and it replenishes the fluids proactively; thus, the intraocular pressure is consistently maintained near the target value. Under such circumstances, stable intraocular pressure and anterior chamber volume values could be acquired. Research on surgical safety, efficiency and results have reported several strengths of the active-fluidics system. It is also advantageous in some complicated cataract surgeries. However, the system is not widely used at present, mainly due to its low penetration rate and high equipment cost. Some of its updates such as the new Active Sentry handpiece showed potential superiority in laboratory studies recently, but there is still further research to be conducted. This article gives an overview of the mechanism and performance of the active-fluidics system, and it is expected to provide clues for future research.

## 1. Introduction

With the aging of people, cataracts have become a major eye disease that is inevitable and severely threatens one’s vision acuity [1]. To date, phacoemulsification (PKE) combined with intraocular lens implantation is still the preferred method for modern cataract surgery [2].

For successful PKE to be completed, the interplay of various systems in a phacoemulsifier is essential [2,3,4]. Among them, the fluidics system is crucial in maintaining the balance of fluid flow and adjusting the volume of the anterior chamber (AC). The evolution and development of the fluidics system coincided with the PKE technique. In as early as 1967, in the first coverage of PKE, a fluid flow device was integrated into the handpiece to flush the AC and cool the phaco tip [5]. Since then, the gravity-fluidics method has been widely used and has become an integral part of the fluidics system in the following decades [6].

More recently, the active-fluidics system (AFS) has come to the attention of ophthalmologists [7,8,9]. It is capable of maintaining intraoperative intraocular pressure (IOP) stability and by virtue of this feature has brought a certain degree of improvement in the safety, efficiency and outcomes of PKE. Here, we summarize and review the studies regarding the AFS in cataract surgery, hoping to better understand the features of this new technique and to provide clues for future research.

## 2. Two Different Fluidics Systems of PKE

Safe and effective fluid irrigation is a prerequisite for PKE, and the fluidics systems currently used in clinical practice include the gravity-fluidics system (GFS) and AFS.

### 2.1. GFS

The GFS is a conventional fluidics system in which the balanced salt solution (BSS) bottle is lifted high and hence possesses gravitational potential energy [10,11]. When the fluid flows into the eye, the irrigation pressure is determined by the height difference between the BSS bottle and the operated eye, that is, a higher bottle equals a higher irrigation pressure. By elevating or lowering the bottle height, the flow and speed of the irrigation could be altered. However, the rate of fluid aspiration is dynamic during PKE, and the relatively permanent irrigation rate in the GFS cannot replenish it in a timely manner, which renders inevitable fluctuating IOP during surgery. In this case, the surgeon needs to adjust the bottle height according to the AC depth [12,13]. For example, the bottle should be raised when the AC becomes shallow. This kind of modulation is delayed, detrimental to the surgical continuity, and requires the surgeon to have considerable experience. At the same time, fluctuating AC and IOP are prone to impair ocular tissues and retinal blood circulation, causing related complications [10,14].

### 2.2. AFS

#### 2.2.1. Characteristics of AFS

Launched in 2013, the AFS allows surgeons to preset a target IOP value and utilizes a soft BSS bag, which is put into the phacoemulsifier as source of irrigation fluid [15,16]. It is paired with a cassette that contains pressure sensors monitoring IOP changes in real time. When the IOP drops below the pre-set pressure, two metal plates will automatically compress the BSS bag to increase irrigation pressure and flow rate [7,15]. Then, the intraocular fluid and AC volume will be rapidly compensated. Meanwhile, when the IOP is higher than the pre-set value, the decompression of plates allows for the regulation of IOP through a reverse process. Due to this flexible modulation, the irrigation fluid can be replenished promptly even in the case of variable aspiration rates, sustaining IOP at the set value and avoiding the detectable fluctuation of the AC volume [17].

At any time during the operation, the operator could reset the target IOP according to different conditions, and a fluent surgery is guaranteed [8,10].

#### 2.2.2. The Influence of Different Settings

Recently, a series of experimental studies on porcine lenses have explored the results of different surgical settings on efficiency and chatter [18]. Efficiency is defined here as the time needed for lens removal, and chatter is defined as the number of lens fragment repulsions from the tip. Based on previous research which showed that the torsional ultrasound (US) mode has superiority over the longitudinal mode in PKE under the GFS, Jensen and colleagues tested it in the AFS [18,19]. By increasing the torsional power from 10% to 100%, the efficiency was linearly increased from 30% to 60% power and then reached a plateau, while the minimum chatter was observed at 60% power. Therefore, 60% torsional power was suggested as the optimized choice. Then, further studies evaluated the influence of continuous, micropulse and long-pulse torsional US on efficiency and chatter events separately [20,21]. However, neither micropulse or long-pulse US was proved to improve the efficiency.

There were also studies focused on longitudinal US in the AFS. Thomson et al. tested longitudinal US power at 40%, 70% and 100% with the torsional power at 0% and considered that 70% power was the most efficient setting among them [22]. After that, Bohner and colleagues conducted a similar experiment in which the torsional power was set at 100% [23]. A linear increase in efficiency was observed as the longitudinal power rose from 20% to 100%, and they recommended 60% longitudinal power as the preferred setting to increase efficiency and avoid chatter. When a comparative study was performed on human eyes, the 100% torsional mode was more efficient than the 60% longitudinal mode in the AFS, though the visual acuity and corneal injuries were comparable between them [24].

Other settings such as the flow rate and vacuum are also factors that affect the efficiency of PKE significantly. A flow rate of 40 mL/min was once determined to be more efficient than 20 mL/min and 60 mL/min [25]. Additionally, a high-vacuum setting of 600 mmHg was proved to be more efficient than 350 mmHg [12]. However, it is worth noting that the optimal choice may vary with different settings of US power, flow rate and vacuum values, and the above outcomes should be interpreted with caution.

## 3. Performance of AFS in PKE

Assessments of safety and efficiency were the most concerns in terms of the evaluation of a new system or the adjustment of the PKE setting preference by surgeons [21]. Research on the performance of the AFS has been accomplished in comparison with the GFS. Due to the heterogeneity between different studies, some dissenting voices exist. An overview of main studies comparing the AFS and GFS is presented in Table 1.

### 3.1. Safety

#### 3.1.1. Anterior Chamber Stability

PKE is performed in the anterior segment of the eye, so a stable AC, which is closely associated with the IOP, is necessary. A proper AC volume can provide enough space for surgery, but a shallow or fluctuating AC could hamper the operation and make the cornea, iris and lens capsule vulnerable [12]. Only when the rates of irrigation and aspiration are balanced can the IOP remain constant, thus forming a gratified AC [7,17]. However, fixed irrigation pressure cannot accommodate the transient aspiration rate during PKE with the GFS, and the IOP tends to decrease with increases in aspiration rate, resulting in variants in the AC volume [10,16]. Nicoli and colleagues monitored IOP in an acrylic test chamber under varying flow conditions with the AFS. Their study showed that the IOP was consistently maintained near the target value when the aspiration rate varied in 15–60 cc/min [10]. Yesilirmak et al. also noticed similar advantages of the AFS in maintaining IOP and AC stability [13]. Flexible irrigation pressure could immediately compensate for variation in fluid flow, adapt quickly to fluctuations in aspiration rate and improve the safety.

Moreover, contrary to the GFS, the IOP in the AFS ascends smoothly at the onset of irrigation thanks to its capability of IOP ramping, which avoids the sudden deepening of AC [10,32]. Yet, quantitative indicators for evaluating the AC volume were absent until the spring eye model was developed [33]. Based on this model, several studies have evaluated aqueous volume losses in different conditions, which are shown in the next section.

Other factors such as the surgical technique used and the choice of handpiece also have an impact. Smaller incisions, needles and sleeves all facilitate protections against outflow. The newer Active Sentry handpiece, which incorporates a pressure sensor, allows for the more accurate, immediate sensing of IOP changes and faster responses to AC volume losses [34,35]. However, no clinical studies are available regarding upgrades on human eyes.

#### 3.1.2. Intraoperative Adverse Events

Commonly occurring adverse events in PKE include occlusion break, posterior capsule (PC) rupture, suspensory ligament injury and so forth. Among them, occlusion break is most closely associated with the fluidics system. Occlusion occurs when the tip is blocked by lens fragments, the iris or a viscoelastic agent during aspiration, and the vacuum in the line and cassette would then rise rapidly [17]. Once the occlusion breaks, the abrupt aspiration of fluid from AC will follow, which is known as a surge. An occlusion break surge can lead to a sudden drop in the IOP, shallow or even collapsed AC, increased risk of PC rupture and accidental damages. It is the origin of many surgical accidents and is intimately associated with surgical safety.

To reduce surge when performing PKE with the GFS, the surgeon’s frequent coping strategy is to elevate the BSS bottle and counteract it with a high irrigation pressure. Studies reported that the amplitude of surge is linearly correlated with the bottle height. For every 60 cm elevation in bottle height, surge is reduced by approximately 40% [11]. However, it is not feasible to set the bottle unlimitedly higher. Meanwhile, a strength of the AFS is that replenishing the fluid in a timely and proactive manner minimizes IOP fluctuations [7,13]. The peristaltic pump system equipped in the AFS controls the fluid displacement or aspiration flow actively and generates vacuum passively [36]. Although the original control of irrigation by sensors at the pump did not allow for a proper increase in irrigation when no flow was detected during occlusion, a vacuum sensor in the phacoemulsifier remedied the defects. Moreover, the new Active Sentry handpiece is an additional surge mitigation feature, which dampens the surge volume demand once the onset of an occlusion break is detected by the pressure sensor [34]. A laboratory study using a transducer box reported that in the range of 200–600 mmHg vacuum settings, the surge in the AFS was smaller than that in the GFS [17]. The difference grew with the increment in vacuum value, and a higher irrigation pressure in the AFS also yielded a smaller surge.

Further laboratory studies that deployed the spring eye model provide more detailed data. Aravena and colleagues evaluated aqueous volume losses due to occlusion breaks at vacuum limits of 200–600 mmHg [36]. Among several commonly used fluidics systems, the AFS had the lowest surge volume and corresponding percent aqueous volume losses. This is certainly strong evidence of better AC stability in the AFS, and this strength is reinforced with the new handpiece. Laboratory studies reported that compared to either the GFS or the plain AFS, the surge volume significantly reduced with Active Sentry handpiece upgrades [34,37]. A subsequent study in rabbits measured the real-time, dynamic IOP fluctuations after occlusion break [35]. The results showed that the AFS combined with the Active Sentry handpiece could achieve lower IOP fluctuations and faster recovery to the target IOP. Yet, there remains a need for more evidence to verify its performance in human eyes.

Additionally, one retrospective study compared the efficacy between the AFS and the GFS in 286 eyes, which reported that the incidence of surges was significantly lower with the AFS than the GFS, and the incidence of PC rupture was lower as well, substantially increasing the safety of PKE [38].

#### 3.1.3. Postoperative Complications

Despite the advances in cataract surgery technology, postoperative complications are inevitable, of which, corneal edema is the most frequent. Exposure to surgical lights, thermal injury from US energy, the collision of nuclear debris and the washout of irrigation fluid are all causes of corneal damage [39]. When the corneal endothelium is damaged, its unique “pump” function will become impaired, which would increase water content in the corneal stroma, lead to corneal edema and result in vision loss [40]. Studies have reported significantly less central corneal thickness (CCT) and corneal edema in the early postoperative period after PKE with AFS compared with GFS [27,38,41]. This advantage of the AFS cannot be achieved without its ability to maintain stable AC and fluid flow [27,40]. The stable AC prevents accidental contact of the cornea with the tip and reduces potential intraocular tissue injuries. The stable fluid flow not only reduces the scouring effect of the turbulent flow on the cornea, but also avoids crashes of the nuclear debris. On top of this, the high surgical efficiency of AFS further reduces the damage from US energy.

The effect of PKE on retinal structure and function has also been the focus of recent research. Increases in retinal thickness (RT) and retinal vessel density (VD) after PKE have been observed, and such changes are closely related to surgical injuries, IOP changes and inflammatory reactions [30,31,42,43,44]. Comparisons between the AFS and GFS on this topic may reveal some potential mechanisms. Studies have reported significant increases in postoperative RT, retinal nerve fiber layer thickness, and ganglion cell layer thickness in patients who underwent PKE with the GFS [9,42]. Their changes in RT lasted 3 months, but RT did not show significant changes in the AFS group. According to OCTA results, the peripapillary VD and retinal superficial VD increased significantly in the GFS group in the early postoperative period and only gradually plateaued at 90 days postoperatively, while the AFS group became stable at 30 days [9,43]. These differences may be attributed to the sudden increase in intraocular irrigation pressure and fluctuations in the IOP when performing surgery with the GFS, which can dampen the retina and optic nerve easily [14,16,45]. Meanwhile, in the AFS, there is no need to maintain AC stability by high irrigation pressure, and a stable controlled intraoperative IOP reduces arterial perfusion pressure variations. All these characteristics can reduce retinal structural damages of surgical origin and contribute to better outcomes.

No significant difference has been yet reported between the AFS and GFS in terms of other surgical complications. Randomized controlled studies reported no significant difference in the incidence of adverse events such as allergic conjunctivitis, ocular congestion, dry eye, and posterior vitreous detachment after PKE with each of the two fluidics systems, and none had other serious complications [7,29,31].

### 3.2. Efficiency

Surgical efficiency is an important aspect to evaluate the working performance of a fluidics system. Indicators for assessing efficiency include: cumulative dissipated energy (CDE), total aspiration time (TAT) and estimated fluid usage (EFU), etc. [7,26]. Jointly used in studies to evaluate efficiency from different perspectives, these parameters could be automatically calculated using software and presented in the panel of a phacoemulsifier. CDE, as the most important item, is the total amount of ultrasound energy consumed during nucleus removal [13,46]. Additionally, a small portion of the released energy can be absorbed by ocular tissues and converted into thermal energy, causing a temperature elevation while generating free radicals. Therefore, the efficient usage of CDE is of great importance, as it will reduce temperature increment, corneal endothelial cell (CEC) damage and inflammation in AC and facilitate early postoperative recovery [46].

Several studies reported that the least CDE, EFU and TAT were consumed when applying the AFS in similar nuclear grades [7,8,9,29]. However, the outcomes of different studies were not entirely consistent, with reports of CDE savings from the AFS ranging from 13.5% to 40% [7,27,29]. There were also reports of no significant difference in the comparison of CDE [31,47]. The reason behind this is that many factors affect the surgery efficiency, including the fluidics system, surgical techniques, needle tips, the vacuum value, and so on. Skilled surgeons can perform the surgery quickly while reducing the CDE and EFU consumption [48]. The balanced tip has also been proven to be more efficient than the Kelman tip [27,49]. Meanwhile, the setting of a high vacuum value is another important aspect in improving the efficiency [12,49].

Due to the high vacuum condition, not only the adsorption of the tip to the nucleus is enhanced but certain fragments can be directly aspirated, thus saving CDE and improving the efficiency [50]. However, owing to the risk of surges, the vacuum value in the GFS is limited. Meanwhile, the ability of the AFS to maintain AC and reduce surges allows the vacuum to be set higher while guaranteeing safety. A prospective clinical study reported a 26.2% reduction in CDE and 17.6% less time taken to emulsify half of the nucleus in the AFS at a high vacuum of 600 mmHg compared to 350 mmHg [12]. Along with the improved efficiency, no AC shallowing was observed in any of the 160 cases, suggesting better safety. All of the above indicates that the advantages of AFS can only be maximized with a combination of skilled surgeons, efficient needles and a high vacuum setting.

### 3.3. Effects

#### 3.3.1. Visual Acuity

Several studies have compared the recovery of visual acuity after PKE with the AFS and GFS, confirming the superiority of the AFS in fostering early postoperative recovery. The postoperative best corrected visual acuity (BCVA) is one of the most important indicators to assess the outcome of cataract surgery, which is closely related to the patient’s primary visual functions and surgical injuries. Some clinical studies demonstrated that BCVA in the early postoperative period (1 day and 1 week) was better when PKE was performed with the AFS than with the GFS [31,47,51]. This advantage did not vanish until 1 month postoperatively [8,38,47]. The better results of early postoperative vision recovery with the AFS may be due to its high efficiency. With less CDE, the inflammatory response and corneal edema after surgery decrease, and they are sure to shorten the recovery period and enable better visual acuity.

In addition, the improvement of visual quality after PKE has also been paid more attention recently. A recent study reported less higher-order aberrations and higher modulation transfer function with the AFS, providing better postoperative visual quality [41]. Reduced visual quality manifestations such as glare and poor night vision caused by higher-order aberrations are associated with altered corneal morphology after cataract surgery. As it reduces corneal damages as well as the impact on visual quality, the AFS is quite valuable in the long run.

#### 3.3.2. Patients’ Subjective Perceptions

To prevent the collapse of AC during PKE with the GFS, surgeons usually elevate the BSS bottle to increase the irrigation pressure and counteract surges [11]. Yet, this can cause discomfort such as eye distention and eye pain, and it is detrimental to patients’ cooperation [10,15]. In addition, a high IOP may induce ocular perfusion reduction and optic nerve damage, even increasing patients’ anxiety and worsening the surgical outcome [45,52]. The maintenance of AC stability and the ability to combat surges provided by AFS can improve surgical safety and efficiency and also allows for a low target IOP to be set (40–50 mmHg). If the target IOP is closer to the physiological status, patients’ discomfort can be greatly reduced and the surgery can be better coordinated [9,15,31].

## 4. AFS in Complicated Conditions

Admittedly, many cataract patients have comorbid ocular or systemic diseases. Theoretically, the strengths of the AFS in terms of surgical safety and efficiency can provide some benefits in complex cataract surgery. However, in different cases, the settings and results are varied.

### 4.1. Corneal Disorders Combined with Cataract

Cataract patients with coexisting keratopathy, such as Fuch’s endothelial dystrophy, age-related CECs dysfunction and keratoplasty history, are commonly found, especially in advanced hospitals. It is essential to obtain satisfactory vision acuity with a healthy, transparent cornea, but endothelial cell loss (ECL) associated with PKE injuries can range from 4% to 25% [53]. Accompanying that loss is the inability of the cornea to maintain its dehydrated state, which can lead to corneal edema and endothelial decompensation. During PKE, the US energy is the most vital and direct contributor to CEC damages [39]. Therefore, better efficiency and less CDE are goals to be pursued relentlessly.

Higher surgical efficiency as well as stable AC reduce the possibility and extent of CEC damages when the AFS is applied to PKE. A retrospective study reported that the incidence of corneal edema after PKE with the AFS was lower, and ECL at 1 month postoperatively was also significantly less than that in the GFS [38]. The results of another randomized controlled study demonstrated the continued loss of CECs at 6 months postoperatively [27]. However, fortunately, lesions of the cornea were less severe in patients who underwent the procedure with the AFS. The conclusions above indicate that the AFS has valuable applicability and a protective effect on CECs, especially for patients combined with corneal endothelial dysfunction or corneal transplantation history, whose CEC barrier function is weak and tolerance towards injury is poor [54,55,56]. However, how good these theoretical advantages actually are needs to be examined in further studies.

Diabetic keratopathy, as one of the ocular complications of diabetes, is characterized by impaired ECE function and corneal nerve alterations in the limbal area, resulting in delayed wound healing and low visual acuity [57]. Meanwhile, cataracts always occur early and progress rapidly in diabetic patients under the influence of hyperglycemia [57,58]. Recent studies showed that CECs of diabetes patients are vulnerable in PKE, particularly in those with poor glycemic control [56,59]. Severe ECL in diabetes may be correlated with CECs’ vulnerability and increased trauma from US energy [60]. Nonetheless, the strengths of the AFS in terms of efficiency and safety may offer a potential turnaround. Previous studies showed that cataracts combined with diabetes required more CDE and resulted in more corneal damages in PKE [60,61]. Meanwhile, a prospective clinical study reported similar surgical efficiency in patients with or without diabetes when PKE was performed with AFS [39]. Although the CCT was higher in the diabetic group 1 day postoperatively, the count of CECs 1 day and 1 month postoperatively did not differ significantly between the diabetics and nondiabetics. This definitely provides a new view on the use of the AFS in cataract patients with diabetes.

### 4.2. Cataracts in Short Eyes and Long Eyes

Eyes with short axial length (AL) are characterized by a shallow AC, narrow angle and thick lens. PKE in patients with short AL is very challenging because of their compacted anterior segment and limited space for maneuvers [62,63,64]. Maintaining a stable AC is crucial there, as a fluctuating IOP, especially a low IOP, greatly increases the risk of choroidal detachment, hemorrhage and PC rupture [15]. With the ability to respond quickly to IOP changes, the AFS provides a suitable operating space, reducing accidents and significantly augmenting safety. Chang et al. reported three cataract patients (five eyes in total) with short AL who successfully underwent PKE with AFS [15]. All patients experienced excellent stability of the AC volume and PC. Only one case (two eyes) was complicated by high IOP and macular edema.

In contrast, when the AL is more than 26 mm or the refraction is greater than −6.00D, this is called high myopia [65]. Cataract patients with high myopia always present with nuclear cataract and capsular bag relaxation [66,67]. The CECs of high myopia are less tolerant to surgical trauma and are liable to be impaired by US energy [68]. In addition, their vitreous gel is often liquefied, which provides less support and leads to fluctuations in the AC volume [65]. In this case, the application of AFS has obvious benefits. For one thing, it can reduce CDE and hence diminish the occurrence of corneal edema. For another, the optional low irrigation pressure minimizes patients’ dissatisfaction and is able to maintain a stable AC, which lessens traction on the zonule and prevents the migration of vitreous gel. However, no high-quality studies have been reported on such patients, which may be the subject of future works.

### 4.3. Cataract with Glaucoma or Vitrectomy

Apart from the anatomical characteristics of short eyes, glaucoma patients usually exhibit a combination of elevated IOP and optic nerve impairment. The defective optic nerve is less resilient to the ischemic and hypoxic environment caused by IOP fluctuations [14,42,69]. Studies showed that in cataract patients suffering from primary angle-closure glaucoma (PACG), PKE with the AFS alleviated the inflammatory congestion of the optic disc, reduced retinal nerve fiber layer edema and offered great protection towards the optic nerve [70]. Moreover, patients with glaucoma require continuous monitoring of the IOP, whereas a postoperative corneal edema directly impacts the accuracy of IOP measurements. In cataract patients with PACG, the adoption of the AFS enabled less corneal edemas and more accurate IOP measurements in the early postoperative period [71]. These accomplishments are significant for identifying postoperative IOP control in glaucoma patients and assessing their conditions promptly.

Vitrectomy has become one of the treatments for many vitreoretinal diseases, whereas a cataract is the most frequent complication [72]. PKE in a vitrectomized eye is risky due to the lack of vitreous support, insufficient zonules and invisible tears of PC, which render it prone to AC fluctuation and PC hyperactivity [73]. Additionally, there is a tendency for reverse pupillary block to induce an abrupt deepening of AC. Should PKE be performed with the GFS, then the bottle height requires prompt reduction to re-establish the AC [74]. Meanwhile, in the AFS the IOP and AC are maintained better, avoiding deep AC and minimizing ocular pain associated with zonule pull. A retrospective study investigated the safety and efficacy of the AFS in cataract patients after vitrectomy [75]. The results showed that all included patients (32 eyes) had stable AC without significant surges. The relatively low target IOP settings (50–60 mmHg) avoided impacts on the fragile optic nerve from transient high IOP, and the high efficiency lessened corneal damages. Despite the small sample size of this study, the value of AFS in such patients was adequately demonstrated.

### 4.4. Cataract with Soft and Dense Nucleus

According to the Emery–Little classification, nuclear scleroses grade II or less are often referred to as “soft” cataracts, while nuclear scleroses grade III or higher are known as “hard” cataracts [76,77]. Both of them are quite challenging in PKE.

Conventional splitting techniques may not work in a soft nucleus given its fragile essence. Additionally, the nucleus can be chopped quickly, rendering PC liable to be aspirated and destroyed. Thus, maintaining the stability of AC is crucial, and this is where the high vacuum and low energy of the AFS fuel the success of PKE [40,78]. The high vacuum allows the nucleus to be sucked tightly and emulsified away from PC, reducing the risk of capsule rupture. Meanwhile, the nucleus leaves the capsule bag in closer proximity to the corneal endothelium, where a lower CDE contributes to less CEC damage. Davison recommended a surgical technique using an optimized “Slow Quadrant Removal” setting to reduce the aspiration pace and prevent PC aspiration in soft cataracts with the AFS [79]. However, conclusions regarding the effectiveness and safety still need to be confirmed in high-quality studies. Additionally, a prospective study presented better efficiency in the AFS compared with the GFS [80]. It was a convincing demonstration of the high vacuum advantage and proof that the AFS could also improve surgical efficiency in soft cataracts.

Meanwhile, in cases of PKE with dense nuclei, emulsification requires higher energy consumption, takes a longer time and requires more irrigation fluids, leaving CECs susceptible to damages [81]. Moreover, since many patients are accompanied with weak zonule, attention should be paid to the stretching injury to the capsular bag [50]. Here, the AFS can maintain a stable AC, reduce the movement of the capsular bag owing to IOP fluctuations and minimize injuries to the zonule [12]. Simultaneously, its capability to efficiently exploit US energy and high vacuum alleviates postoperative corneal edema with better surgical results. It was demonstrated that in dense nuclear cataract patients, the AFS significantly improved surgical efficiency, reduced corneal edema, and achieved faster visual recovery [51]. The results from another prospective research also showed that PKE with the AFS resulted in a 28% reduction in CDE and significantly better BCVA than the GFS 1 month postoperatively in dense nuclear cataract patients [8]. In addition, the presence of a hard nucleus increases the risk of occlusion during surgery, while the resistance to surges in the AFS certainly further adds to the safety.

## 5. Summary and Future Directions

The launch of the AFS was an important milestone in the history of fluidics systems, which marked a new phase of active and controlled intraoperative fluid perfusion. The strengths of the AFS are mutually reinforcing: its ability to maintain AC stability allows for better lens fragments followability, which reduces invalid maneuvers and thus improves the efficiency [41]; the result of saving EFU and TAT cannot be achieved without the automatic regulation of fluid flow and pressure [7,27,29]; the reduction in CDE, EFU and TAT improves the efficiency and lessens injuries from US energy and hydrodynamic effects, contributing to faster vision recovery.

The AFS is a technical breakthrough on the basis of the GFS, which guarantees the success of many complex cataract surgeries. However, due to its newness, the system is not widely used, and the research is inadequate; hence, many of its theoretical advantages have not been proven. For example, the irrigation pressure in the AFS was reported to be linearly correlated with surgical efficiency in a laboratory study, and increasing irrigation pressure alone could improve efficiency significantly [16]. However, no reasonable range of IOP settings was recommend, and no clinical trial was conducted to validate them. Furthermore, whether the protection of retinal blood flow by the AFS can reduce the incidence of macular edema, especially in diabetic patients, remains to be verified.

It is worth mentioning that the AFS has also undergone a revolution. The upgraded Active Sentry handpiece, which detects IOP via sensors, enables the mitigation of a surge response faster. But studies with large cohorts and long follow ups are still badly needed. Femtosecond laser-assisted cataract surgery (FLACS) is becoming popular, as it was reported to be more efficient [82]. This kind of strength has also been observed when both FLACS and conventional PKE were conducted with the AFS, and the balanced tip further enlarged it [82,83].

Currently, the utilization of the AFS is still limited in some regions for several reasons. First, there is a lack of clinical studies, leaving many potential strengths of it unexplored. Second, although the AFS has advantages in many complex cataract surgeries, it is not “one size fits all”. The AFS is available for PKE only but is not a substitute for extracapsular cataract extraction in some cases such as lens subluxation. Third, the superiority of the AFS is based on the experience and technique of surgeons, and there is a learning curve for the appropriate operation setting and skills [7,8,29]. In addition, the high cost of surgical consumables is an undeniable factor, especially in some developing countries [28]. At present, PKE with the GFS is still the mainstay in many regions, and it is well-qualified for most cataract surgeries. Despite the strengths of the AFS, the GFS can work wonders provided the operating team, including surgeons and assistants, understand the mechanism of it and use it properly.

The development and utilization of new devices is the certain consequence of technological progress. With thorough research on the clinical application of the AFS in the years ahead, its practical value will be further analyzed and reported. The future development and discussion of fluidics systems will become a hot spot and will certainly start a new chapter of cataract surgery.

## Figures and Tables

**Table 1 jcm-12-00611-t001:** Overview of main studies comparing AFS and GFS.

Author	Study Type	System	Tips	Parameters	Sample	AFS vs. GFS
Chen et al. [26]	Cohort	AFS */GFS (Infiniti)	Bal/Kel	CDE	2077	Less CDE
Solomon et al. [7]	RCT	AFS/GFS (Infiniti)	Bal/Kel	CDE, AFU, AT	200	Improved surgical efficiency
Yesilirmak et al. [13]	Cohort	AFS/GFS (Infiniti)/FLACS + AFS/FLACS + GFS (Infiniti)	Bal/Kel/Bal/Kel	CDE	570	Less CDE
Malik et al. [27]	RCT	GFS (Infiniti)/AFS/AFS	Kel/Kel/Bal/	CDE, EFU, TAT, ECDCCT, CV, 6A	126	Improved surgical efficiency, better corneal preservation
Oh et al. [8]	Cohort	AFS/GFS (Infiniti)	Bal/Kel	CDE, CDVA	412	Less CDE, better CDVA
Huang et al. [28]	Cohort	AFS/GFS (Infiniti)/GFS (WhiteStar Signature)/GFS (Stellaris Vision Enhancement systems)	Bal/NA/NA/NA	CDE, surgery duration, BCVA, AE	150	Improved surgical efficacy, less AE and highest cost
Gonzalez-Salinas et al. [29]	Cohort	AFS/GFS (Infiniti)	Bal/Kel	CDE, AFU, AT, ECD, CCT	164	Improved surgical efficiency
Zhao et al. [9]	RCT	AFS/GFS (Centurion)	Bal/Bal	CDE, EFU, TAT, RVD, MT	50	Improved surgical efficiency, less disturbance of RVD and MT
Liu et al. [30]	Cohort	AFS/GFS (Centurion)	NA	VD in peripapillary and parafoveal	40	Improved surgical efficiency and retinal vasculature protection
Luo et al. [31]	RCT	AFS/GFS (Centurion)	Bal/Bal	CDE, EFU, TAT, ECD, CCT, CV, 6A, IOP, BCVA, RVD, RT, AE et al.	107	Improved surgical efficiency, effects, safety and patients’ subjective perceptions

* All the AFS is used in Centurion. AE—adverse event, AFS—active-fluidics system, AFU—aspiration fluid used, AT—aspiration time, Bal—balanced, BCVA—best corrected visual acuity, CCT—central corneal thickness, CDE—cumulative dissipated energy, CDVA—corrected distance visual acuity, CV—coefficient of variance, ECD—endothelial cell density, EFU—estimated fluid usage, FLACS—femtosecond laser-assisted cataract surgery, GFS—gravity-fluidics system, IOP—intraocular pressure, Kel—Kelman, MT—macular thickness, RCT—randomized controlled trial, RT—retinal thickness, RVD—retinal vessel density, TAT—total aspiration time, 6A—percentage of hexagonal endothelial cells.

## Data Availability

No new data were created or analyzed in this study. Data sharing is not applicable to this article.

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
