# Peer review of "Application of the Active-Fluidics System in Phacoemulsification: A Review"

_jcm, 2023, doi:10.3390/jcm12020611_

Round 1
Reviewer 1 Report
This review is comprehensive and well organized. However the mechanism of the active fluidics system is not enough detailed. It should be mentioned that the original control of irrigation by laser sensors at the pump did not allow proper increase of irrigation when no flow was detected during occlusion, and that vacuum sensors at the handpiece were introduced to overcome this problem.
Author Response
Thank you very much for your meticulous work as the description was not detailed enough. We have added the relevant content based on your suggestion, which is located on Page 5 Line150-157. Now it reads “The peristaltic pump system AFS equipped control the fluid displacement or aspiration flow actively and generate vacuum passively [30]. Although the original control of irrigation by sensors at the pump did not allow proper increase of irrigation when no flow was detected during occlusion, vacuum sensor in the phacoemulsifier remedy the defects. Moreover, the new Active Sentry handpiece is an additional surge mitigation feature, which dampens the surge volume demand once the onset of an occlusion break is detected by the pressure sensor [28].”
Reviewer 2 Report
The authors have done a commendable job of bringing together the beauty and advantages of Active Fluidics Systems(AFS) while pointing out harshly the negatives of Gravitational Fluidics Systems(GFS). But as an operating surgeon from a developing economy, I would like to bring out the fact that financial considerations are an important part of surgical decisions and infrastructure building. GFS has been in use since the early times and with the correct approach towards each case, GFS can work wonders provided the context is well understood by the operating team including surgeons and assistants. Nonetheless, this article provides a comprehensive overview of the various benefits of the AFS and sounds very convincing to the readers.
Author Response
Thank you very much for the kind reminder. We have revised the statement in Page 10 Line 425-429, and now it reads “Besides, the high cost of surgical consumables is an undeniable factor, especially in some developing countries [83]. Currently, PKE with GFS is still the mainstream in many regions, and it is well qualified for most cataract surgeries. Despite the strengths of AFS, GFS can work wonders provided the operating team including surgeons and assistants understand the mechanism of it and use it properly.”
Reviewer 3 Report
This review is a well-written manuscript about fluidics in phaco tecnique.
References are updated, introduction and comments about gravity-fluidics (GFS) or active-fluid system (AFS) are well-organized and well-presented for the most of eye phaco surgeons and also for eye clinicians. Different cases of PKE have been taken into account, developing both GFS and AFS. The innovation of AFS is undoubtable, but real-life in developing countries makes GFS still used in phaco surgery.
Author Response
Thank you very much for this great comment. The relevant contents have been revised in Page 10 Line 425-429, and now it reads “Besides, the high cost of surgical consumables is an undeniable factor, especially in some developing countries [83]. Currently, PKE with GFS is still the mainstream in many regions, and it is well qualified for most cataract surgeries. Despite the strengths of AFS, GFS can work wonders provided the operating team including surgeons and assistants understand the mechanism of it and use it properly.”
Reviewer 4 Report
The subject addressed is relevant for ophthalmology, because it highlights the advantages of this method compared to GFS, the effectiveness and safety for the patient regarding the favorable evolution in the postoperative period.
The paper analyzes the results of several studies on the advantages of AFS, data that are very important for the training and encouragement of ophthalmologists in its use.
The formulated conclusions are in accordance with the analized results, based on the proposed study objectives. The references are recent and relevant to the information presented. The table is relevant and present the data accurately, being easy to interpret and understand it.
Author Response
We thank reviewer 4 for spending the precious time to review this manuscript and for recognizing our work.